# Does 2x2 airplane passenger contact tracing for infectious respiratory pathogens work? A systematic review of the evidence

Anna C. Rafferty[1,2☯], Kelly Bofkin[3,4☯], Whitney Hughes[3☯], Sara Souter[3,4‡],
Ian Hosegood[3‡], Robyn N. Hall[2‡], Luis Furuya-Kanamori[5‡], Bette Liu[6‡], Michael Drane[7‡],
Toby Regan[8‡], Molly Halder[8‡], Catherine Kelaher[2‡], Martyn D. Kirk[1,2‡]*

1 National Centre for Epidemiology and Population Health, The Australian National University, Canberra,
Australian Capital Territory, Australia, 2 National Incident Centre, The Australian Government Department of
Health, Canberra, Australian Capital Territory, Australia, 3 Qantas Airways Limited, Mascot, New South
Wales, Australia, 4 Virgin Australia Airlines, South Brisbane, Queensland, Australia, 5 UQ Centre for Clinical
Research, Faculty of Medicine, The University of Queensland, Herston, Queensland, Australia, 6 School of
Population Health, University of New South Wales, Kensington, New South Wales, Australia, 7 Air New
Zealand, Auckland, New Zealand, 8 New Zealand Ministry of Health, Wellington, New Zealand

☯ These authors contributed equally to this work.
‡ SS, IH, RNH, LFK, BL, MD, TR, MH, CK and MDK also contributed equally to this work.
* martyn.kirk@anu.edu.au

LUXEMBOURG

**Data Availability Statement:** All relevant data are
within the paper and its Supporting information
files.

## Abstract

We critically appraised the literature regarding in-flight transmission of a range of respiratory infections to provide an evidence base for public health policies for contact tracing passengers, given the limited pathogen-specific data for SARS-CoV-2 currently available. Using PubMed, Web of Science, and other databases including preprints, we systematically reviewed evidence of in-flight transmission of infectious respiratory illnesses. A meta-analysis was conducted where total numbers of persons on board a specific flight was known, to calculate a pooled Attack Rate (AR) for a range of pathogens. The quality of the evidence provided was assessed using a bias assessment tool developed for in-flight transmission investigations of influenza which was modelled on the PRISMA statement and the Newcastle-Ottawa scale. We identified 103 publications detailing 165 flight investigations. Overall, 43.7% (72/165) of investigations provided evidence for in-flight transmission. H1N1 influenza A virus had the highest reported pooled attack rate per 100 persons (AR = 1.17), followed by SARS-CoV-2 (AR = 0.54) and SARS-CoV (AR = 0.32), *Mycobacterium tuberculosis* (TB, AR = 0.25), and measles virus (AR = 0.09). There was high heterogeneity in estimates between studies, except for TB. Of the 72 investigations that provided evidence for in-flight transmission, 27 investigations were assessed as having a high level of evidence, 23 as medium, and 22 as low. One third of the investigations that reported on proximity of cases showed transmission occurring beyond the 2x2 seating area. We suggest that for emerging pathogens, in the absence of pathogen-specific evidence, the 2x2 system should not be used for contact tracing. Instead, alternate contact tracing protocols and close contact definitions for enclosed areas, such as the same

**Funding:** ACR is funded by the Master of Philosophy (Applied Epidemiology) Scholarship at Australian National University. MDK is supported by a National Health and Medical Research Council fellowship (APP1145997). The funders had no role in study design, data collection and analysis, decision to publish, or preparation of the manuscript.

**Competing interests:** Authors KB, WH, SS, IH, and MD have read the journal's policy and the authors of this manuscript have the following competing interests: These authors are all employees of Australian and New Zealand airlines.

cabin on an aircraft or other forms of transport, should be considered as part of a whole of journey approach.

## Introduction

International travel has played a major role in the rapid global spread of SARS-CoV-2, the pathogen responsible for COVID-19 [1–3]. The initial response to the pandemic involved restrictions on international travel, which has impacted airlines, commercial aviation, tourism and associated industries. The air travel industry has progressively put in place a variety of interventions to prevent infections occurring before, during and after flying. These interventions include: pre-testing of passengers, physical distancing, enhanced hygiene and cleaning within the aircraft, having passengers and crew wearing masks, leaving middle seats free, and regular screening and testing of crew [4].

Transmission of infectious respiratory pathogens in an aircraft setting is complex. Some of the factors that influence transmission include infectiousness of the agent, timing and severity of a passenger's illness, the nature of ventilation and filtration, space limitations, and the proximity and duration of interactions between passengers [5–7]. In Australia, the current public health approach is to contact trace passengers seated in the two rows in front of the case, the row of the case and the two rows behind the case, across the width of the fuselage, which is referred to as 2x2 contact tracing. This contact tracing strategy is based on the landmark outbreak investigation of Severe Acute Respiratory Syndrome (SARS) by Olsen and colleagues in 2003 [8]. Despite this, there have been reports of transmission events reported beyond this seating configuration [9]. Furthermore, physical distancing in-flight may not be sustainable or commercially viable as the demand for travel increases. The evidence for in-flight transmission can also be confounded by interactions elsewhere during the journey, for example in the airport terminal or in transit to or from the airport [10].

Previous systematic reviews have evaluated the risk of in-flight transmission of various infectious respiratory pathogens, including influenza virus (seasonal and H1N1), *Mycobacterium tuberculosis* (TB) and measles virus [5, 6, 11, 12]. These previous reviews have contributed to the evidence, current policy decisions and frameworks that health agencies are using in the COVID-19 pandemic. More recently a number of SARS-CoV-2 specific reviews have been published, which assess mitigation strategies or aim to estimate attack rates on passenger aircraft, both with conservative estimates from public data and estimates from industry [13–16].

In this review, we critically appraise the literature regarding in-flight transmission of a range of respiratory infections to provide an evidence base for policy given the limited pathogen-specific data for SARS-CoV-2 currently available.

## Methods

In this review, we sought to identify and aggregate the evidence for transmission of infectious respiratory illnesses on aircraft and assess the utility of 2x2 contact tracing. The protocol of this review was registered with the international prospective register of systematic reviews (PROSPERO CRD42020191261) [17].

### Search strategy

We searched PubMed, Web of Science and the Cochrane library for articles containing information on the transmission of respiratory illness on an aircraft or in-flight. MedRxiv and

BioRxiv were searched for preprints regarding flight associated transmission of SARS-CoV-2. There was no language restriction on the search, however articles that were not published in English or did not have an adequate translation available were excluded. Further search details can be found in S1 Appendix. The International Air Transport Association (IATA), the Collaborative Arrangement for the Prevention and Management of Public Health Events in Civil Aviation (CAPSCA), WHO, ICAO, EASA, CDC and ECDC databases were searched to include all relevant studies and industry documents. We conducted the search on 20 May, 2021.

Two reviewers screened titles and abstracts, with a third reviewer resolving conflicts for inclusion or exclusion. For full text screening, two reviewers reviewed each article, with conflicts resolved through discussion between reviewers. We conducted additional searches of outbreak investigation reports in the CDC, ECDC and CDI databases, to identify outbreak investigations where flight -associated and in-flight contact tracing was undertaken as part of an investigation but was not reported on separately. We included these where there was evidence of contact tracing and reported outcomes.

We included studies detailing investigations into in-flight transmission. We excluded studies that were based on modelling and simulation, along with previous systematic reviews. Additionally, during the data extraction phase we excluded articles where there was inadequate data presented in the study and/or a lack of investigation of affected flights.

Data were extracted in duplicate from each article to ensure consistency. Data extracted for analyses included the pathogen, number of passengers, number of persons on a flight (passengers and crew), number of infective cases, number of secondary cases, the index case definition, the secondary case definition, length of flight, proximity of secondary cases to the index case/s, contact tracing strategy, timeliness of contact tracing, and alternative exposures addressed. Extracted data are summarised in S1 Table.

During the data extraction process, references of all included articles were manually searched for additional relevant articles. We estimated attack rates for flights where the total number of persons (or susceptible persons) aboard an aircraft were reported, regardless of the contact tracing strategy employed. The pooled attack rates by pathogen were estimated using the inverse variance heterogeneity model [18]. The double arcsine square root transformation was applied to stabilise the variance; results were reported after back-transformation for ease of interpretation [19]. All tests were two-tailed, and $P \leq 0.05$ was deemed statistically significant. Pooled analyses were conducted using MetaXL version 5.3 (EpiGear International, Sunrise Beach, Queensland, Australia).

We used a bias assessment tool, developed by Leitmeyer *et al.* for influenza investigations, to evaluate the level of evidence for transmission aboard aircraft [6]. The bias assessment tool was modelled on the PRISMA statement and the Newcastle-Ottawa scale. Articles were assessed on the strength of evidence of each investigation and categorised as low, medium, or high evidence, based on factors relevant to contact tracing, such as methodology, timeliness and outcomes. Bias assessment was conducted in duplicate.

We modified the bias assessment tool for TB investigations (changing timeframes from weeks to months) to account for the difference in transmission dynamics and the timeframe for contact tracing appropriate for this pathogen (S2 Appendix). TB studies were also complicated by the extended latency period and corresponding delays in investigations. TB investigations commonly involved multiple flights, rather than being conducted on an individual flight basis as is typical of more acute pathogens such as SARS-CoV-2, SARS-CoV, Middle East Respiratory Syndrome (MERS) virus or H1N1 influenza A virus, which have pandemic potential and rapid transmission [20, 21].

## Results

### Study characteristics

We screened 425 titles, resulting in the inclusion of 103 articles in our review of which, 60 of these articles were identified during secondary searches (S1 Appendix) [22]. In total, there were 165 flight investigations detailed in 103 articles that we included in our review (Fig 1), after accounting for duplicate publications of investigations into H1N1, MERS, and SARS-CoV-2 (Fig 1).

Respiratory pathogens included in our review included SARS-CoV (n = 5) [8, 23–26], MERS virus (n = 12) [27–38], TB (n = 20) [20, 21, 39–56], measles virus (n = 27) [57–83],

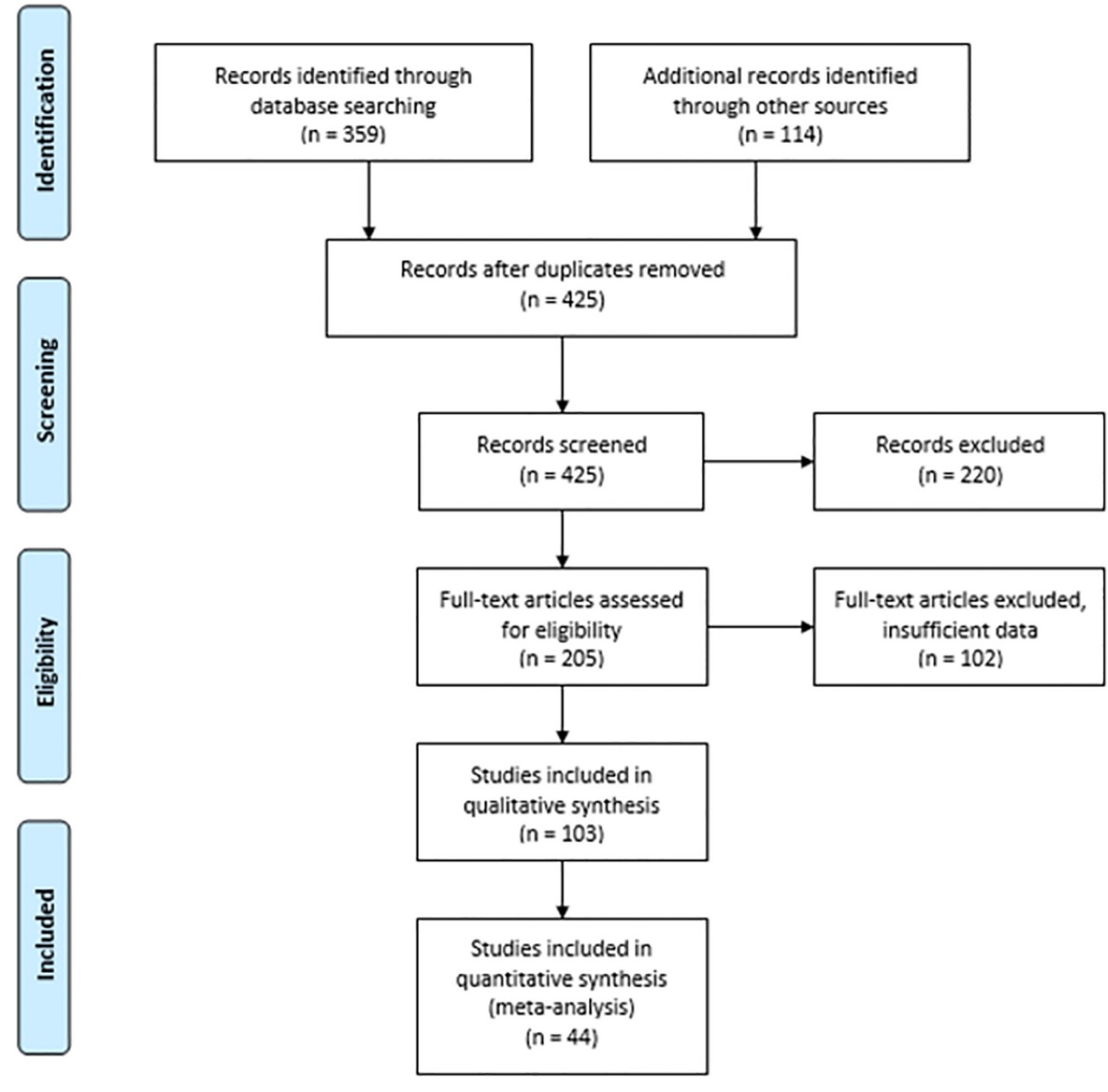

**Fig 1. PRISMA 2009 flow diagram** [22].

mumps virus (n = 2) [84, 85], rubella virus (n = 1) [86], *Corynebacterium diphtheriae* (n = 1) [56], H1N1 influenza A virus (n = 10) [12, 87–95], seasonal influenza virus or influenza like illness (ILI; n = 2) [96, 97], *Neisseria meningitidis* (n = 3) [98–100], and SARS-CoV-2 (n = 20) [1, 3, 10, 101–117].

The two articles on MERS virus reported on the same flights and investigations in the United States and the United Kingdom, with complementary data presented in each paper [27, 28]. We have only counted the four flights presented in each of these papers once but have treated them as two investigations based on two different index cases. We identified other articles on H1N1 influenza A virus and measles virus that reported duplicate investigations, which were counted once for the purposes of data analysis [57, 63, 64, 78, 87, 91]. We identified two articles on the same flight outbreak for SARS-CoV-2 with contradictory conclusions about the number of secondary cases due to in-flight transmission. For the purposes this review, we included the investigation with greater epidemiological evidence and robust discussion around pre and post flight exposures [107]. We included an unpublished Australian report detailing several investigations into flight-associated transmission of SARS-CoV-2 [117].

Index case classification was based on laboratory confirmation in 89.1% (147/165) of investigations. Of the remaining 18 investigations, either clinical presentation was used for index case classification or it was not reported. Investigators used various definitions for secondary cases due to the different pathogen characteristics and public health protocols. In total, 84.8% (140/165) of investigations used pathogen-specific testing criteria to define secondary cases, 5.5% (9/165) of investigations used clinical characteristics, while the remaining 9.7% (16/165) of investigations did not provide sufficient detail on how secondary cases were characterised.

Of the investigations that reported in-flight transmission, 97.2% (70/72) considered alternative exposures when determining whether in-flight transmission may have occurred.

## Transmission and proximity

Overall, 43.7% (72/165) of investigations provided evidence for in-flight transmission of respiratory illness based on their epidemiological investigations. Of these, 21 were for SARS-CoV-2 [1, 3, 10, 101, 103, 105, 107, 108, 110–113, 115, 117], 13 for H1N1 influenza A virus [4, 12, 87–91, 93–95], 22 for measles virus [58–66, 69–71, 73–77, 79–83], two for seasonal influenza virus and ILI [96, 97], four for SARS-CoV [8, 25, 26], seven for TB [39, 43, 45, 47, 51, 118], one for mumps virus [85], and two for *N. meningitidis* [99, 100].

We calculated attack rates for 5 pathogens in 59 investigations (44 articles) where investigators provided data on the total numbers of passengers on board. H1N1 influenza A virus had the highest pooled attack rate per 100 persons exposed (AR = 1.17, 82/6456, 95% CI; 0.0000–0.82) (Fig 2), followed by SARS-CoV-2 (AR = 0.54, 63/7260, 95% CI; 0.00–1.71) (Fig 3) and SARS-CoV (AR = 0.32, 25/2835, 95% CI; 0.00–1.17) (S1 Fig), TB (AR = 0.25, 8/3212, 95% CI; 0.07–0.49) (S2 Fig), and measles virus (AR = 0.09, 17/11918, 95% CI; 0.00–0.82) (S3 Fig).

There was considerable heterogeneity amongst investigations included in our review, ($I2>50\%$), except for TB. We present the pooled attack rate meta-analysis for all pathogens in Table 1, with forest plots for the remaining pathogens in S1–S3 Figs.

Of the investigations where in-flight transmission was documented, 63.9% (46/72) reported on the proximity of secondary cases to the index case/s. In 31.1% (14/46) of investigations, transmission was reported to have occurred exclusively within two rows of an index case [1, 12, 69, 70, 95, 101, 107, 110, 117]. For the remaining 32 investigations, transmission occurred beyond this 2x2 area. Overall, in the 46 investigations where proximity to the index case was reported on, 48.7% (94/193) of reported secondary cases occurred outside of the 2x2 seating area around the index case.

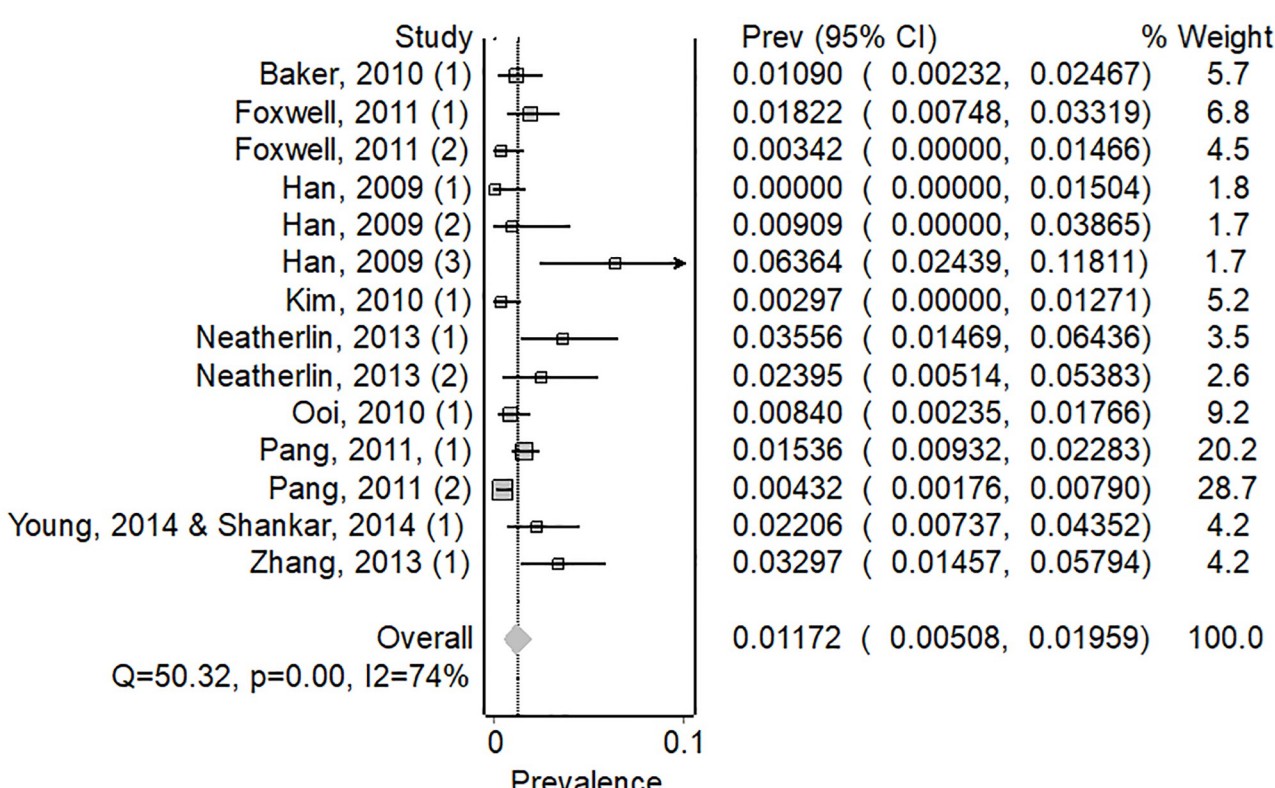

**Fig 2. Forest plot showing weighted pooled attack rates for H1N1.**

Of the 32 investigations reporting transmission outside the 2x2 zone, six were for SARS-CoV-2 [10, 103, 105, 111, 112, 117], three for TB [43, 47, 118], seven for H1N1 influenza A virus [87–92], 11 for measles virus [59–61, 64, 66, 71, 73, 75–77, 80], three for SARS-CoV [8, 25, 26], one for ILI [97], and one for meningococcal meningitis [99].

For SARS-CoV-2, investigators reported proximity in 71.4% (15/21) of investigations and 80.7% (46/57) of secondary cases occurred within the 2x2 seating area surrounding an index case. Proximity was reported in 77.0% (10/13) of H1N1 investigations with 51.1% (23/45) of secondary cases occurring within the 2x2 zone. Proximity was reported in 13 of 22 measles investigations with 27.5% (11/40) of secondary cases occurring within the 2x2 zone. Proximity was reported in 3 of 4 SARS-CoV investigations with 24% (6/25) of secondary cases occurring within the 2x2 zone. Proximity was reported in 3 of 7 TB investigations, with 40% (4/10) of secondary cases occurring within the 2x2 zone. For the single ILI and influenza investigation reporting proximity, 40% (9/15) of secondary cases occurred within the 2x2 zone. For the single meningococcal investigation reporting proximity none of the secondary cases (n = 1) occurred within the 2x2 area around an index case.

## Assessment of the evidence and bias

We evaluated that 46 investigations had a high level of evidence, 71 had medium, and 48 had a low level of evidence demonstrating the occurrence or absence of in-flight transmission. The median evidence score of 5 (range: -1–9). Eight investigations achieved the highest score of 9, with four of these being investigations for SARS-CoV-2. These are detailed in S1 Table.

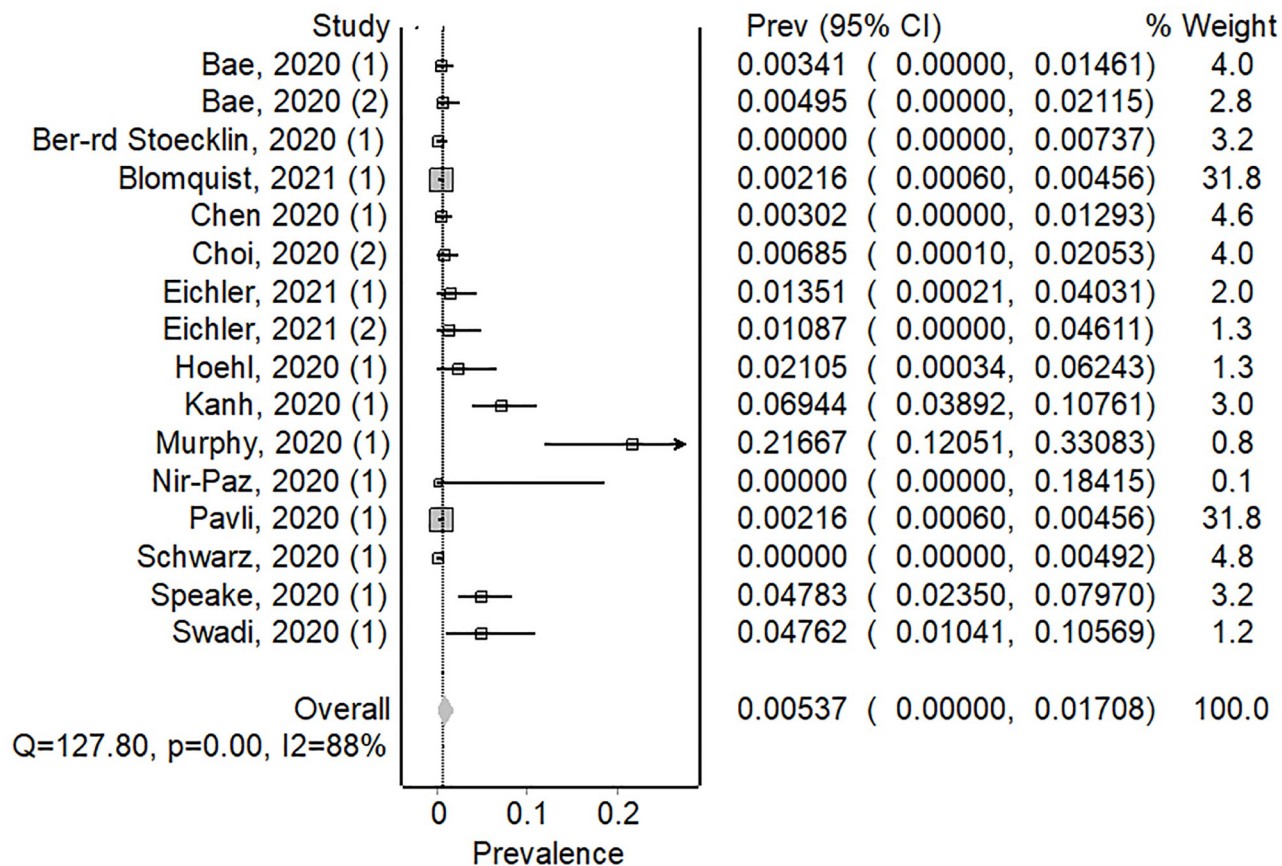

**Fig 3. Forest plot showing weighted pooled attack rates for SARS-CoV-2.**

**Table 1. Estimated attack rates of in-flight transmission of selected respiratory pathogens and number of investigations, reported index cases, and reported secondary cases, by pathogen.**

| Pathogen | Number of articles in meta-analysis (n = 44) | Pooled attack rate (per 100 persons exposed) | 95% Confidence Interval | I²% | Number of unique investigations in review (n = 165) | Number of index cases[#] | Number of secondary cases |
|---|---|---|---|---|---|---|---|
| Measles virus | 11 | 0.09 | 0.00–0.82 | 80.7 | 31 | 228 | 62 |
| SARS-CoV-2 | 14 | 0.54 | 0.00–1.71 | 88.3 | 29 | 204 | 87 |
| H1N1 influenza A virus | 10 | 1.17 | 0.51–1.96 | 74.2 | 14 | 36 | 82 |
| SARS-CoV | 3 | 0.32 | 0.00–1.17 | 87.6 | 19 | 24 | 26 |
| *Mycobacterium tuberculosis* | 6 | 0.25 | 0.07–0.49 | 20.0 | 51 | 667 | 31 |
| MERS virus | - | - | - | - | 12 | 14 | 0 |
| Mumps virus | - | - | - | - | 2 | 13 | 2 |
| Rubella virus | - | - | - | - | 1 | 1 | 0 |
| *Neisseria meningitidis* | - | - | - | - | 3 | 3 | 3 |
| Influenza virus (ILI) | - | - | - | - | 2 | 2 | 53 |
| *Corynebacterium diphtheriae* | - | - | - | - | 1 | 1 | 0 |

[#]Number of index cases not always reported

Table 2. Evidence assessment by pathogen, where in-flight transmission has been reported (n = 72).

| Pathogen | Number of investigations and evidence level | | |
|---|---|---|---|
| | Low | Medium | High |
| SARS-CoV-2 | 3 | 8 | 10 |
| H1N1 influenza A virus | 2 | 4 | 7 |
| *Mycobacterium tuberculosis* | 0 | 3 | 4 |
| Measles virus | 14 | 3 | 5 |
| SARS-CoV | 0 | 4 | 0 |
| Mumps virus | 1 | 0 | 0 |
| *Neisseria meningitidis* | 2 | 0 | 0 |
| Seasonal influenza virus & ILI | 0 | 1 | 1 |
| Total | 22 | 23 | 27 |

It was determined that there were 72 investigations providing evidence that in-flight transmission occurred and of these, 27 were assessed as having a high level of evidence, 23 as medium, and 22 as low. Breakdown by pathogen is detailed in Table 2.

## Discussion

We found strong evidence for in-flight transmission of a range of respiratory pathogens, particularly for SARS-CoV-2. We found that where proximity was known, more than half of respiratory pathogen transmission events, occurred outside of the standard arrangement that public health uses to contact trace a 2x2 seating area around an infected passenger. Pre and post flight information can be utilised within investigations into potential transmission in-flight, but this does not appear to be common practice. Integration of this information and may shift the focus towards flight-associated transmission as the primary outcome of interest rather than the current focus on the inflight period.

All studies in our review focussed on the potential for transmission to occur on the aircraft during the flight. However, a flight is not a singular event. The logistically complex nature of air travel leads to passenger and crew interactions at all stages of the air travel process, i.e., pre-flight, in-flight, and post-flight. Potential transmission, for example, could occur at the airport terminal (check-in, baggage, customs and immigration, in a transit lounge or at the gate) or on public transport to or from the airport and has been demonstrated previously in the case of measles [76]. When analysing seating maps, distant infections beyond the traditional 2x2 zone may be explained by these alternative pathways. Instances of in-flight transmission may be considered to have most likely occurred within the cabin environment, but this cannot be conclusively determined. Genomic evidence can link index and secondary infections who travelled on the same flight for many pathogens, including SARS-CoV-2, but cannot definitively determine the circumstances of transmission [10].

Although measles virus is highly infectious, high vaccination rates internationally and the availability of post exposure prophylaxis is likely reflected in the low attack rate observed in our review [64]. Treatment and post exposure measures are also applicable to TB, with a similarly low attack rate (AR = 0.25). Due to the rapid spread of SARS-CoV-2 in 2020, investigations into in-flight transmission were likely to be more robust as a result of higher public awareness and intense public health response and a vaccine naïve population during the period included in this review. However, this also increases the potential for publication and confirmation bias.

Just under half of the secondary cases in investigations that reported on proximity showed transmission occurring beyond the 2x2 area. However, this does not mean that public health agencies should not use the 2x2 area for contact tracing. We suggest that, where circumstances require, public health agencies may wish 7to extend tracing beyond the 2x2 zone. These circumstances could include flights in regions with a pathogen elimination strategy, where a novel or highly infectious pathogen has emerged, or where there are multiple infectious passengers on the flight. In these scenarios, public health agencies could use alternate contact tracing protocols and definitions of close contacts for enclosed areas, such as the same cabin on an aircraft or other forms of transport. This is particularly true for highly infectious variants of SARS-CoV-2, such as the Delta variant [119].

We observed considerable heterogeneity in attack rates between studies included in the meta-analysis. This is likely due to the stochastic nature of outbreaks, differences in outbreak investigation and logistical challenges presented by contact tracing on domestic and international flights. In addition, it is likely that the many instances where infectious cases transmitted to one or no other passengers were not published. Our review does not indicate how commonly cases did not transmit outside the 2x2 zone, but does indicate that it occurs from time to time. In addition, many published reports of investigations involved multiple infectious cases on the same flight, which would lead to a higher infection pressure.

We recommend that where public health investigators are alerted to infectious cases travelling on a flight, they should conduct a risk assessment of the characteristics of the flight including relevant infection controls utilised, as well as that of the airport transit locations when investigating transmission of highly transmissible or high-consequence pathogens. Public health authorities should conduct potential outbreak investigations in conjunction with airlines and airports where practical, noting the current challenges with contact tracing surrounding international travel. Additionally, where there is more than one index case on a flight, the 2x2 area may be inappropriate and a cabin or whole-of-plane contact tracing strategy should be undertaken after adequate risk assessment and consideration of available resources, in line with current public health strategies in their jurisdiction.

## Limitations

This review is subject to several limitations including the massive emerging literature on SARS-CoV-2; difficulties in defining secondary cases as having been acquired in-flight; the heterogeneity of included studies; and the variable quality of included studies. Within the first year of the World Health Organization declaring the SARS-CoV-2 pandemic, more than 200,000 journal articles and preprints on SARS-CoV-2 were published [120]. The high volume of SARS-CoV-2 publications may well result in studies being missed due to the confines of the search strategy and the dynamic nature of publications. We did search CDC, ECDC and IATA databases, however, we are mindful that there would have been many instances where flight contact tracing was undertaken as part of outbreak investigations but never published.

Our review included investigations into SARS-CoV-2 prior to roll-out of vaccines in countries, widespread immunity due to either natural infection or vaccination, and the emergence of variants of concern with known increases in transmissibility and reduction in incubation period. As such, some of our findings may not be applicable to the current circulating strains of SARS-CoV-2 and public health control measures in place at the time of publication, but our findings do highlight that the 2x2 system of contact tracing itself is not specific enough for contact tracing management of SARS-CoV-2 in all circumstances.

We were unable to conduct a meta-analysis on 2x2 contact tracing itself, only on the attack rates on flights. It is important to consider that investigative findings may have been driven by

the definition of who is classified as a contact. However, our descriptive analysis of 2x2 contact tracing does provide evidence that cases of infectious disease do occur outside of 2x2 contact tracing zones, which may require different management approaches.

We assessed the level of evidence using a bias and assessment tool that we adapted from Leitmeyer *et al.* [6] This allowed us to assess each investigation to determine the evidence level. We used the tool to assess the methods used in each investigation, which identified a high degree of heterogeneity. The methodological issues identified highlight the weakness of many investigations within this review, contributing to the overall median rating of medium evidence level in this review. The high proportion of investigations with low and medium evidence scores is attributable to the majority of investigations using the narrow 2x2 contact tracing strategies, the time between flight and commencement of contact tracing (often attributed to delays in clinical presentation and subsequent diagnosis), and the incomplete nature of contact tracing. Furthermore, it is not always clear within the papers reviewed what provision has been made to handle such confounders during an outbreak investigation. All of these factors would lead to potentially missed secondary cases, introducing bias into the studies.

Contact tracing investigations from air travel are complicated and are often incomplete, although this has changed with greater use of electronic data for tracing. Close contact follow up is particularly difficult for international travel. The ability of any one public health unit to undertake a thorough contact tracing investigation to determine if in-flight transmission has occurred is limited due to multiple jurisdictions, international travel and lack of contact details for passengers who are in transit. This is highlighted in a number of multi-national investigations included in this review, where the same flights or cases were reported by different jurisdictions, such as a MERS virus outbreak that was investigated in the UK and the US [27, 28] and an international flight with a H1N1 investigation that was investigated by multiple jurisdictions [87, 91].

Asymptomatic infections may also be underreported as testing may only be conducted on symptomatic individuals. For SARS-CoV-2, many countries, including Australia and New Zealand, have required all incoming passengers to undertake mandatory quarantine in hotels or other purpose-built facilities where they are routinely tested therefore providing opportunities for more complete follow-up [121].

Case studies of in-flight outbreaks detail secondary cases that are identified with an epidemiological link and investigated further. As case studies do not assist in quantifying the risk of in-flight transmission, they should not solely be relied on as an evidence base, but rather provide an exploratory tool to prompt further study. The relevance of case studies decreases after the initial stages of pathogen emergence once robust epidemiological investigations have occurred. Furthermore, contact tracing investigations are undertaken for the purpose of implementing public health measures, rather than for research. During an epidemic or pandemic, contact tracing capacity may be overwhelmed and robust tracing may not occur. Consequently, we cannot determine the frequency of transmission outside the 2x2 zone of seating, other than it does occur. Public health agencies may choose to use wider contact tracing strategies depending on their local disease control priorities and resource constraints.

Investigations using routine surveillance data are only appropriate for notifiable diseases resulting in health outcomes that require presentation to medical services. Otherwise, investigations are likely underreporting the occurrence of potential in-flight transmission. However, for the pathogen SARS-CoV-2, some countries have high levels of testing, including asymptomatic testing, meaning that there is high ascertainment of cases. Therefore, the use of surveillance systems may be adequate to evaluate potential in-flight transmission events for SARS-CoV-2 but is limited for other pathogens. In particular, the use of 2x2 contact tracing may reinforce the belief that it is effective if no tracing and testing occurs outside this zone.

Surveillance systems are likely to underrepresent cases for notifiable diseases that are less common, cause mild illness, have no treatment, are relatively rare, or do not require a laboratory test for treatment and management. Retrospective analysis of surveillance data is not recommended as an investigative approach for public health action due to potential underreporting, inherent in surveillance systems, and lack of timeliness.

## Conclusion

In our review, we determined that air travel related transmission of pathogens responsible for many respiratory illnesses occurred outside of the standard 2x2 area for contact tracing. However, we have found overall that this evidence was only of a medium level of quality and raises questions about reliance upon a single 2x2 contact metric. In contrast, our findings indicate that in certain circumstances, a whole-of-flight or whole-of-journey approach may be necessary for contact tracing persons infected with emerging pathogens until pathogen-specific transmission dynamics are understood or most travellers have immunity. Consideration of the utility of contact tracing of this nature in reducing transmission within a country or community is vital to preserve resources. Case studies of outbreaks provide valuable initial insights into the in-flight transmission of SARS-CoV-2. The low level of evidence is attributable to under detection and subsequent underreporting of cases across public health units, multinational investigations and lack of airline cooperation rather than a reflection of the public health investigators methods. It is important to recognise that accounts of in-flight or flight associated transmission are uncommon in medical literature, and that the publication of case studies do not reflect the majority of flights where transmission does not occur.

The entire air travel process, from travel to an airport to departing the terminal at the end destination, needs to be considered in terms of infection control and interaction during public health investigations. This style of multilayered, approach has been recommended by the International Civil Aviation Organisation and IATA, and should be considered by public health authorities when investigating infectious respiratory diseases in the context of flight [122]. This multilayered approach will help protect public health and enable containment of infectious respiratory pathogens as international travel resumes.

## Supporting information

**S1 Appendix. Search strategy.**
(DOCX)

**S2 Appendix. Quality assessment tool, adapted from Leitmeyer, et al. [6].**
(DOCX)

**S1 Table. Supplementary data and evidence assessment.**
(DOCX)

**S1 Fig. Forest plot showing weighted pooled attack rates for SARS-CoV.**
(TIF)

**S2 Fig. Forest plot showing weighted pooled attack rates for *Mycobacterium tuberculosis*.**
(TIF)

**S3 Fig. Forest plot showing weighted pooled attack rates for measles.**
(TIF)

**S1 Checklist. PRISMA 2020 checklist.**
(DOCX)

## Acknowledgments

We acknowledge and thank our Australian and New Zealand health department and airline colleagues for their assistance with this review and participating in discussion regarding in-flight transmission.

## Author Contributions

**Conceptualization:** Anna C. Rafferty, Kelly Bofkin, Whitney Hughes, Sara Souter, Ian Hosegood, Robyn N. Hall, Luis Furuya-Kanamori, Bette Liu, Toby Regan, Catherine Kelaher.

**Data curation:** Anna C. Rafferty, Kelly Bofkin, Whitney Hughes.

**Formal analysis:** Anna C. Rafferty, Whitney Hughes, Luis Furuya-Kanamori.

**Investigation:** Anna C. Rafferty, Kelly Bofkin.

**Methodology:** Anna C. Rafferty, Robyn N. Hall, Luis Furuya-Kanamori, Martyn D. Kirk.

**Project administration:** Anna C. Rafferty.

**Supervision:** Martyn D. Kirk.

**Writing – original draft:** Anna C. Rafferty, Whitney Hughes.

**Writing – review & editing:** Anna C. Rafferty, Kelly Bofkin, Whitney Hughes, Sara Souter, Ian Hosegood, Robyn N. Hall, Luis Furuya-Kanamori, Bette Liu, Michael Drane, Toby Regan, Molly Halder, Catherine Kelaher, Martyn D. Kirk.

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
