## [Decision Letter · Decision Letter 0]

20 Apr 2022

PONE-D-22-03741Does 2x2 airplane passenger contact tracing for infectious respiratory pathogens work? A systematic review of the evidence.PLOS ONE

Dear Dr. Kirk,

Thank you for submitting your manuscript to PLOS ONE. After careful consideration, we feel that it has merit but does not fully meet PLOS ONE’s publication criteria as it currently stands. Therefore, we invite you to submit a revised version of the manuscript that addresses the points raised during the review process.

Both reviewers agree that your manuscript would benefit from changes. Please address all of their suggestions.==============================

We look forward to receiving your revised manuscript.

Kind regards,

Joël Mossong

Academic Editor

PLOS ONE

Journal Requirements:

“ACR is funded by the Master of Philosophy (Applied Epidemiology) Scholarship at Australian National University. MDK is supported by a National Health and Medical Research Council fellowship (APP1145997).”

“ACR is funded by the Master of Philosophy (Applied Epidemiology) Scholarship at Australian National University. MDK is supported by a National Health and Medical Research Council fellowship (APP1145997).

5. Please ensure that you refer to Figure 2, 3, 4, 5 and 6 in your text as, if accepted, production will need this reference to link the reader to the figure.

Reviewers' comments:

Reviewer's Responses to Questions

**Comments to the Author**

1. Is the manuscript technically sound, and do the data support the conclusions?

Reviewer #1: Yes

Reviewer #2: Yes

2. Has the statistical analysis been performed appropriately and rigorously? 

Reviewer #1: Yes

Reviewer #2: I Don't Know

3. Have the authors made all data underlying the findings in their manuscript fully available?

Reviewer #1: Yes

Reviewer #2: Yes

4. Is the manuscript presented in an intelligible fashion and written in standard English?

Reviewer #1: Yes

Reviewer #2: Yes

5. Review Comments to the Author

Reviewer #1: I read it with great interest, but I have raised several concerns.

#1. In my common sense, it is known that 2*2 contact tracing in flight was accepted as a well-established theory after publising the landmark previoust study in NEJM 2003. I hope to have a more specific discussion on this.

Olsen SJ, Chang HL, Cheung TY, Tang AF, Fisk TL, Ooi SP, Kuo HW, Jiang DD, Chen KT, Lando J, Hsu KH, Chen TJ, Dowell SF. Transmission of the severe acute respiratory syndrome on aircraft. N Engl J Med. 2003 Dec 18;349(25):2416-22. doi: 10.1056/NEJMoa031349. PMID: 14681507.

#2. In abstract method, please describe the assessing tool of evidence level for Plos One readers.

#3.Since the author did not analyze meta-statistics for 2*2 contact tracing, the main results may be over-interpreted.

Reviewer #2: REVIEW - PONE-D-22-03741

Title: Does 2x2 airplane passenger contact tracing for infectious respiratory pathogens work? A systematic review of the evidence.

PLOS ONE

1. Summary of the research and your overall impression

General comment:

This study addresses the relevant question of which literature is available to date of in-flight transmissions of respiratory infections (in the context of limited data available for SARS-CoV-2) for informed public health policies and recommendations for passenger contact tracing.

The method (Systematic Literature Review) is well chosen to answer the research question. The review workflow was well conducted from the registration of the review on PROSPERO 2020 CRD42020191261 in the beginning, the use of the respective PRISMA guidelines/ checklist, the utilised databases and other sources (incl. relevant international organisations/ authorities, search for relevant grey literature, snowballing) as well as the review process itself (e.g. screening, definition of inclusion/ exclusion criteria, data extraction). I do not have the expertise to consider the statistics in detail, but the methods of analysis given seem appropriate (AR, pooled AR/meta-analysis).

The authors are aware of the limitations of their study and have taken these into account in their conclusions.

Recommended course of action:

The identified evidence can be useful for future decision making and recommendations. It would therefore be helpful to publish the article. However, this paper could benefit from some clarifications in the results and discussion section (see part 2).

Suggested improvements:

Publish with minor revision

2. Discussion of specific areas for improvement

Major Issues

Results: Study characteristics: page 8, line 144-147 and figure 1:

The authors should clarify the following section to avoid confusion. The numbers in the text of the manuscript and the flowchart do not correspond. Figure 1 shows “studies included in qualitative synthesis n=102”. Line 144 refers to 103 enclosed articles. If applicable, add “number finally included studies in meta-analysis” to the text to link flowchart and results (line 185 and table 1).

Results: page 13: line 210-213:

The authors should check and revise (if applicable) the numbers in this passage. The authors refer to 32 investigations where transmission outside the 2x2 zone was reported, but the sum of the investigations listed per pathogen does not add up to 32.

Results: page 14, Assessment of the evidence and bias, line 227-233:

The authors should clarify the following section to avoid confusion. Here, reference is made to studies that demonstrate in-flight transmission or provide evidence of in-flight transmission, but with different denominators [in line 227-228: n=165 (sum of 46, 71 and 48) and in line 231: n=72]. The difference between the denominators is not clear to the reader.

"We assessed that 46 studies had a high level of evidence ... demonstrating in-flight transmission." (see line 227) and "Of the 72 studies that provided evidence of in-flight transmission, ..." (see line 231).

My understanding of the data presented is that the first denominator refers to the "number of unique investigations in the review", i.e. all 165 flight investigations (see line 146). The second denominator refers to investigations providing evidence of in-flight transmission of respiratory illness (see line 178-179).

Discussion: page 16: line 264-266:

The authors should check and clarify the following sentence: “Just under half of the investigations that reported on proximity showed transmission occurring beyond the 2x2 area.”

From my understanding of the data presented, there are 46 investigations that reported on proximity, of which 32 investigations reported transmission outside the 2x2 zone (see results page 13). That would result in almost 70% (32/46) and would be more than half of the investigations.

Perhaps the authors confused investigations and secondary cases in this summary of results, see page 13, line 208: “…, 48.7% (94/193) of reported secondary cases occurred outside of the 2x2 seating area around the index case.”

Minor Issues

Introduction: page 4, line 71-73:

Please add reference for this statement: “The evidence for in-flight transmission can also be confounded by interactions elsewhere during the journey, for example in the airport terminal or in transit to or from the airport.”

Introduction: page 4, line 73-74:

In my opinion the following sentence is not part of the introduction, rather part of the results or discussion. If desired, move the statement to the respective section of the paper.

“It is not always clear within the papers reviewed what provision has been made to handle such confounders during an outbreak investigation.”

Methods: page 7, line 130-133:

The authors may indicate how many reviewers assessed the risk of bias (see PRISMA Checklist Item 11). Likewise, two reviewers as for the data screening and extraction process?

“Articles were assessed on the strength of evidence of each investigation and categorised as low, medium, or high evidence, based on factors relevant to contact tracing, such as methodology,

timeliness and outcomes.”

Methods: page 7, line 136-141:

Please add reference for the following paragraph: “Tuberculosis studies were also complicated by the extended latency period and corresponding delays in investigations. Tuberculosis investigations commonly involved multiple flights, rather than being conducted on an individual flight basis as is typical of more acute pathogens such as SARS-CoV-2, SARS-CoV, Middle East Respiratory Syndrome (MERS) or H1N1, which have pandemic potential and rapid transmission.”

Results: page 8-9, line 164-166:

Please add the reference for the publication mentioned. This is presumably reference #118.

“We included an unpublished Australian report detailing several investigations 165 into flight-associated transmission of SARS-CoV-2.”

Results: page 11, table 1:

Please add the denominator in column 1 "Number of articles in the meta-analysis" (n=44) and column 5 "Number of individual studies in the review" (n=165) to better link the manuscript text and the data reported in the table.

Results: page 14, table 2:

Please add the denominator (n=72) in the caption to better link the manuscript text and table.

Discussion: page 15, line 237-240:

The following finding is a repetition, please see result section line 208-209. Consider rephrasing this finding without repeating the numbers.

“We found that 48.7% (94/193) of all respiratory pathogen transmission events, where proximity was reported, occurred outside of the standard arrangement that public health uses to contact trace a 2x2 seating area around an infected passenger”

Discussion: page 16, line 277-278:

I am not a native English speaker, but the authors should consider rephrasing the following sentence regarding the clause ".... to one or fewer persons ..." to make the meaning clearer: “In addition, it is likely that the many instances infectious cases transmitted to one or fewer persons were not published.” The wording seems misleading, persons count as a whole, and less than one person is "no" person. Perhaps the authors mean "one or only a few persons", but then I think it would be useful to quantify the "few".

Ref. 14: The format of the title seems wrong: “Probability and Estimated Risk of SARS-CoV-2 Transmission in the Air Travel System: A Systemic Review and Meta-Analysis” doi: 10.1101/2021.04.08.21255171; doi: 10.1016/j.tm

6. PLOS authors have the option to publish the peer review history of their article (what does this mean?). If published, this will include your full peer review and any attached files.

Reviewer #1: No

Reviewer #2: No

---

## [Author Response · Author response to Decision Letter 0]

12 Oct 2022

Response to reviewers

Reviewer 1

#1: I read it with great interest, but I have raised several concerns. #1. In my common sense, it is known that 2*2 contact tracing in flight was accepted as a well-established theory after publising the landmark previoust study in NEJM 2003. I hope to have a more specific discussion on this. Olsen SJ, Chang HL, Cheung TY, Tang AF, Fisk TL, Ooi SP, Kuo HW, Jiang DD, Chen KT, Lando J, Hsu KH, Chen TJ, Dowell SF. Transmission of the severe acute respiratory syndrome on aircraft. N Engl J Med. 2003 Dec 18;349(25):2416-22. doi: 10.1056/NEJMoa031349. PMID: 14681507.

Response: We agree that this study is critical importance, and is considered a landmark study for coronavirus transmission. The importance of this piece of work has been emphasised in the revised manuscript at line 71. 

#2. In abstract method, please describe the assessing tool of evidence level for Plos One readers.

Response: The description of the evidence assessment has been updated in the abstract to include additional detail at line 38.

#3.Since the author did not analyze meta-statistics for 2*2 contact tracing, the main results may be over-interpreted.

Response: Thank you for this comment. Meta analysis was not considered in this context due to potential biases. We have included this in the revised manuscript as a limitation at line 341.

Reviewer 2: REVIEW - PONE-D-22-03741 Title: Does 2x2 airplane passenger contact tracing for infectious respiratory pathogens work? A systematic review of the evidence. PLOS ONE 1. Summary of the research and your overall impression 

General comment: This study addresses the relevant question of which literature is available to date of in-flight transmissions of respiratory infections (in the context of limited data available for SARS-CoV-2) for informed public health policies and recommendations for passenger contact tracing. The method (Systematic Literature Review) is well chosen to answer the research question. The review workflow was well conducted from the registration of the review on PROSPERO 2020 CRD42020191261 in the beginning, the use of the respective PRISMA guidelines/ checklist, the utilised databases and other sources (incl. relevant international organisations/ authorities, search for relevant grey literature, snowballing) as well as the review process itself (e.g. screening, definition of inclusion/ exclusion criteria, data extraction). I do not have the expertise to consider the statistics in detail, but the methods of analysis given seem appropriate (AR, pooled AR/meta-analysis). The authors are aware of the limitations of their study and have taken these into account in their conclusions. Recommended course of action: The identified evidence can be useful for future decision making and recommendations. It would therefore be helpful to publish the article. However, this paper could benefit from some clarifications in the results and discussion section (see part 2). Suggested improvements: Publish with minor revision 2. Discussion of specific areas for improvement 

Major Issues Results: Study characteristics: page 8, line 144-147 and figure 1:

#1: The authors should clarify the following section to avoid confusion. The numbers in the text of the manuscript and the flowchart do not correspond. Figure 1 shows “studies included in qualitative synthesis n=102”. Line 144 refers to 103 enclosed articles. If applicable, add “number finally included studies in meta-analysis” to the text to link flowchart and results (line 185 and table 1).

Response: Thank you, these were typographical errors after a final revision and have been corrected. 

#2: Results: page 13: line 210-213: The authors should check and revise (if applicable) the numbers in this passage. The authors refer to 32 investigations where transmission outside the 2x2 zone was reported, but the sum of the investigations listed per pathogen does not add up to 32.

Response: Thank you, the reviewer is correct. An article for SARS-CoV had been missed but had been referenced.

#3: Results: page 14, Assessment of the evidence and bias, line 227-233: The authors should clarify the following section to avoid confusion. Here, reference is made to studies that demonstrate in-flight transmission or provide evidence of in-flight transmission, but with different denominators [in line 227-228: n=165 (sum of 46, 71 and 48) and in line 231: n=72]. The difference between the denominators is not clear to the reader. "We assessed that 46 studies had a high level of evidence ... demonstrating in-flight transmission." (see line 227) and "Of the 72 studies that provided evidence of in-flight transmission, ..." (see line 231). My understanding of the data presented is that the first denominator refers to the "number of unique investigations in the review", i.e. all 165 flight investigations (see line 146). The second denominator refers to investigations providing evidence of in-flight transmission of respiratory illness (see line 178-179).

Response: Thank you, we have made changes to the revised manuscript to clarify this. 

#4: Discussion: page 16: line 264-266: The authors should check and clarify the following sentence: “Just under half of the investigations that reported on proximity showed transmission occurring beyond the 2x2 area.” From my understanding of the data presented, there are 46 investigations that reported on proximity, of which 32 investigations reported transmission outside the 2x2 zone (see results page 13). That would result in almost 70% (32/46) and would be more than half of the investigations. Perhaps the authors confused investigations and secondary cases in this summary of results, see page 13, line 208: “…, 48.7% (94/193) of reported secondary cases occurred outside of the 2x2 seating area around the index case.”

Response: Thank you for this comment. This has been rectified in the revised manuscript.

Minor Issues 

#5: Introduction: page 4, line 71-73: Please add reference for this statement: “The evidence for in-flight transmission can also be confounded by interactions elsewhere during the journey, for example in the airport terminal or in transit to or from the airport.” Introduction: page 4, line 73-74: In my opinion the following sentence is not part of the introduction, rather part of the results or discussion. If desired, move the statement to the respective section of the paper. “It is not always clear within the papers reviewed what provision has been made to handle such confounders during an outbreak investigation.”

Response: This has been moved to the discussion, as suggested by the reviewer.

#6: Methods: page 7, line 130-133: The authors may indicate how many reviewers assessed the risk of bias (see PRISMA Checklist Item 11). Likewise, two reviewers as for the data screening and extraction process? “Articles were assessed on the strength of evidence of each investigation and categorised as low, medium, or high evidence, based on factors relevant to contact tracing, such as methodology, timeliness and outcomes.”

Response: We have clarified this in the manuscript that the bias assessment process was undertaken in duplicate. 

#7: Methods: page 7, line 136-141: Please add reference for the following paragraph: “Tuberculosis studies were also complicated by the extended latency period and corresponding delays in investigations. Tuberculosis investigations commonly involved multiple flights, rather than being conducted on an individual flight basis as is typical of more acute pathogens such as SARS-CoV-2, SARS-CoV, Middle East Respiratory Syndrome (MERS) or H1N1, which have pandemic potential and rapid transmission.”

Response: Thank you, the appropriate reference has been added to support this claim, 

#8: Results: page 8-9, line 164-166: Please add the reference for the publication mentioned. This is presumably reference #118. “We included an unpublished Australian report detailing several investigations 165 into flight-associated transmission of SARS-CoV-2.”

Response: This is correct and has been included.

#9: Results: page 11, table 1: Please add the denominator in column 1 "Number of articles in the meta-analysis" (n=44) and column 5 "Number of individual studies in the review" (n=165) to better link the manuscript text and the data reported in the table.

Response: The denominators have been added as per the reviewers’ recommendations.

#10: Results: page 14, table 2: Please add the denominator (n=72) in the caption to better link the manuscript text and table.

Response: The denominators have been added as per the reviewers’ recommendations. 

#11: Discussion: page 15, line 237-240: The following finding is a repetition, please see result section line 208-209. Consider rephrasing this finding without repeating the numbers. “We found that 48.7% (94/193) of all respiratory pathogen transmission events, where proximity was reported, occurred outside of the standard arrangement that public health uses to contact trace a 2x2 seating area around an infected passenger”

Response: This has been reworded to avoid duplication in the revised manuscript 

#12: Discussion: page 16, line 277-278: I am not a native English speaker, but the authors should consider rephrasing the following sentence regarding the clause ".... to one or fewer persons ..." to make the meaning clearer: “In addition, it is likely that the many instances infectious cases transmitted to one or fewer persons were not published.” The wording seems misleading, persons count as a whole, and less than one person is "no" person. Perhaps the authors mean "one or only a few persons", but then I think it would be useful to quantify the "few".

Response: This is correct and has been revised to reflect the totality of one person.

#13: Ref. 14: The format of the title seems wrong: “Probability and Estimated Risk of SARS-CoV-2 Transmission in the Air Travel System: A Systemic Review and Meta-Analysis” doi: 10.1101/2021.04.08.21255171; doi: 10.1016/j.tm

Response: This title is unfortunately correct, published as a pre-print by an industry company, using the term “systemic” to be included in search results.

---

## [Decision Letter · Decision Letter 1]

28 Nov 2022

PONE-D-22-03741R1Does 2x2 airplane passenger contact tracing for infectious respiratory pathogens work? A systematic review of the evidence.PLOS ONE

Dear Dr. Kirk,

Thank you for submitting your manuscript to PLOS ONE. After careful consideration, we feel that it has merit but does not fully meet PLOS ONE’s publication criteria as it currently stands. Therefore, we invite you to submit a revised version of the manuscript that addresses the points raised during the review process.

One reviewer still spotted some minor work to be done. Please address this before resubmitting.==============================

We look forward to receiving your revised manuscript.

Kind regards,

Joël Mossong, PhD

Academic Editor

PLOS ONE

Journal Requirements:

Reviewers' comments:

Reviewer's Responses to Questions

**Comments to the Author**

1. If the authors have adequately addressed your comments raised in a previous round of review and you feel that this manuscript is now acceptable for publication, you may indicate that here to bypass the “Comments to the Author” section, enter your conflict of interest statement in the “Confidential to Editor” section, and submit your "Accept" recommendation.

Reviewer #2: All comments have been addressed

2. Is the manuscript technically sound, and do the data support the conclusions?

Reviewer #2: Yes

3. Has the statistical analysis been performed appropriately and rigorously? 

Reviewer #2: I Don't Know

4. Have the authors made all data underlying the findings in their manuscript fully available?

Reviewer #2: Yes

5. Is the manuscript presented in an intelligible fashion and written in standard English?

Reviewer #2: No

6. Review Comments to the Author

Reviewer #2: Thank you for the thorough revision.

Nevertheless, I have 2 minimal comments:

1) Maybe I don't have the right version of the revised manuscript, but the reference to my comment #6 is still missing (please see p. 7, line 137-142).

2) Line288-289: "In addition, it is likely that the many instances where infectious cases transmitted to one no other passengers were not published." If necessary, the author could check this sentence again, as it is still not entirely clear. Perhaps an "or" is missing.

Thank you for your efforts.

7. PLOS authors have the option to publish the peer review history of their article (what does this mean?). If published, this will include your full peer review and any attached files.

Reviewer #2: No

---

## [Author Response · Author response to Decision Letter 1]

17 Jan 2023

Reviewer #2: Reviewer #2: Thank you for the thorough revision. Nevertheless, I have 2 minimal comments:

1) Maybe I don't have the right version of the revised manuscript, but the reference to my comment #6 is still missing (please see p. 7, line 137-142).

Response: Thanks very much. This has been corrected.

2) Line288-289: "In addition, it is likely that the many instances where infectious cases transmitted to one no other passengers were not published." If necessary, the author could check this sentence again, as it is still not entirely clear. Perhaps an "or" is missing.

Response: Thanks very much. This has been corrected.

---

## [Editor Report · Decision Letter 2]

19 Jan 2023

Does 2x2 airplane passenger contact tracing for infectious respiratory pathogens work? A systematic review of the evidence.

PONE-D-22-03741R2

Dear Dr. Kirk,

We’re pleased to inform you that your manuscript has been judged scientifically suitable for publication and will be formally accepted for publication once it meets all outstanding technical requirements.

Kind regards,

Joel Mossong, PhD

Academic Editor

PLOS ONE
---

## [Editor Report · Acceptance letter]

24 Jan 2023

PONE-D-22-03741R2 

Does 2x2 airplane passenger contact tracing for infectious respiratory pathogens work? A systematic review of the evidence. 

Dear Dr. Kirk:

I'm pleased to inform you that your manuscript has been deemed suitable for publication in PLOS ONE. Congratulations! Your manuscript is now with our production department. 

Kind regards, 

on behalf of

Dr. Joel Mossong 

Academic Editor

PLOS ONE